# Development and Validation of a Predictive Tool for Postpartum Hemorrhage after Vaginal Delivery: A Prospective Cohort Study

**DOI:** 10.3390/biology12010054

**Published:** 2022-12-29

**Authors:** Line Bihan, Emmanuel Nowak, François Anouilh, Christophe Tremouilhac, Philippe Merviel, Cécile Tromeur, Sara Robin, Guillaume Drugmanne, Liana Le Roux, Francis Couturaud, Emmanuelle Le Moigne, Jean-François Abgrall, Brigitte Pan-Petesch, Claire de Moreuil

**Affiliations:** 1CIC1412, INSERM, 29200 Brest, France; 2UMR1304, INSERM, GETBO, Université de Bretagne Occidentale, CHRU de Brest, 29200 Brest, France; 3Ecole de Sage-Femmes, UFR Santé, 29200 Brest, France; 4Service de Gynécologie Obstétrique, CHU Brest, 29200 Brest, France; 5Département de Médecine Vasculaire, Médecine Interne et Pneumologie, CHU Brest, 29200 Brest, France; 6CIC-RB Ressources Biologiques (UF 0827), CHU Brest, 29200 Brest, France; 7Centre de Traitement de L’hémophilie, Hématologie, CHU Brest, 29200 Brest, France

**Keywords:** postpartum hemorrhage, vaginal delivery, multiple imputation, bootstrap, predictive model

## Abstract

**Simple Summary:**

Postpartum hemorrhage is a major health issue, affecting pregnant women world-wide. In this study, we derived and validated a robust predictive model identifying women at risk of postpartum hemorrhage after vaginal delivery. We first used clinical and biological data from a prospective cohort of 2742 pregnant women with vaginal delivery at Brest University Hospital (France) between April 2013 and May 2015. We determined then the parameters independently associated with an increased risk of PPH (pre-eclampsia, antepartum bleeding, multiple pregnancy, labor duration ≥ 8 h, macrosomia, episiotomy, platelet count < 150 Giga/L and aPTT ratio ≥ 1.1). Afterwards, we built a predictive score with these parameters, ranging from 0 to 10. Finally, we validated this score on an independent prospective cohort of 3061 vaginal deliveries. This score has the potential to improve the care of pregnant women and to take preventive actions on them.

**Abstract:**

Postpartum hemorrhage (PPH) is one of the leading causes of maternal morbidity worldwide. This study aimed to develop and validate a predictive model for PPH after vaginal deliveries, based on routinely available clinical and biological data. The derivation monocentric cohort included pregnant women with vaginal delivery at Brest University Hospital (France) between April 2013 and May 2015. Immediate PPH was defined as a blood loss of ≥500 mL in the first 24 h after delivery and measured with a graduated collector bag. A logistic model, using a combination of multiple imputation and variable selection with bootstrap, was used to construct a predictive model and a score for PPH. An external validation was performed on a prospective cohort of women who delivered between 2015 and 2019 at Brest University Hospital. Among 2742 deliveries, PPH occurred in 141 (5.1%) women. Eight factors were independently associated with PPH: pre-eclampsia (aOR 6.25, 95% CI 2.35–16.65), antepartum bleeding (aOR 2.36, 95% CI 1.43–3.91), multiple pregnancy (aOR 3.24, 95% CI 1.52–6.92), labor duration ≥ 8 h (aOR 1.81, 95% CI 1.20–2.73), macrosomia (aOR 2.33, 95% CI 1.36–4.00), episiotomy (aOR 2.02, 95% CI 1.40–2.93), platelet count < 150 Giga/L (aOR 2.59, 95% CI 1.47–4.55) and aPTT ratio ≥ 1.1 (aOR 2.01, 95% CI 1.25–3.23). The derived predictive score, ranging from 0 to 10 (woman at risk if score ≥ 1), demonstrated a good discriminant power (AUROC 0.69; 95% CI 0.65–0.74) and calibration. The external validation cohort was composed of 3061 vaginal deliveries. The predictive score on this independent cohort showed an acceptable ability to discriminate (AUROC 0.66; 95% CI 0.62–0.70). We derived and validated a robust predictive model identifying women at risk for PPH using in-depth statistical methodology. This score has the potential to improve the care of pregnant women and to take preventive actions on them.

## 1. Introduction

Postpartum hemorrhage (PPH) remains the most common complication of childbirth and leads to significant maternal morbidity and mortality. Therefore, PPH affects around 5–10% of deliveries worldwide [1].

A few studies have attempted to build predictive models and tools for PPH management by obstetricians, anesthetists and midwives. However, most of them have limited predictive capacities [2,3], have not been validated on an independent cohort of pregnant women [2,3] or do not use a consensual definition for PPH [4]. Finally, despite good knowledge of PPH risk factors, there is currently no predictive tool for PPH in clinical practice, as reported by Neary et al. in their recent systematic review [5]. It is therefore important to establish a generalizable predictive strategy on PPH in order to reduce PPH incidence and severity by means of preventive action implementation.

Based on a prospective “HPP-IPF” study conducted at Brest University Hospital, several clinical parameters (pre-eclampsia, multiple pregnancy, assisted reproduction, bleeding during pregnancy, post-term delivery, obesity and episiotomy) and biological parameters measured upon entry into the delivery room (platelet count, fibrinogen and activated partial thromboplastin time (aPTT) ratio) were found to be associated with PPH [6].

In the present work, the first objective was to build a predictive model for PPH after vaginal delivery using an in-depth statistical methodology, considering the issues of missing data and over-optimization of the model’s performance [7]. A methodology combining multiple imputation and a selection of variables with bootstrap was used.

The second objective was then to externally validate this predictive model for PPH after vaginal delivery in order to develop an easy-to-use tool for clinicians.

## 2. Materials and Methods

### 2.1. Study Design and Participants

The derivation cohort was a monocenter prospective study called “Study of Biological Determinants of Bleeding Postpartum (HPP-IPF)” [6]. This study, conducted at Brest University Hospital between April 2013 and May 2015, aimed to determine the role of biological determinants in the prediction of PPH (NCT02884804). All pregnant women admitted to the Obstetrics unit for childbirth who did not object to participating were included in this study. Women were included upon entry into the delivery room.

The external validation cohort was a subgroup of an independent multicenter prospective study called “Hemorrhages and Thromboembolic Venous Disease of the Postpartum (HEMOTHEPP)” including women who gave birth between 2015 and 2019 (NCT02443610). Only the women who gave birth vaginally at Brest University Hospital were included in the validation cohort.

For the present work, only women who gave birth vaginally after 21 weeks of gestation were analyzed in the two studies. 

### 2.2. Data Collection

The PPH outcome was defined as a blood loss ≥ 500 mL in the first 24 h after vaginal delivery, according to the World Health Organization [8]. Blood loss was measured with a graduated collector bag used routinely after vaginal delivery.

Clinical and biological data were prospectively collected by midwives, obstetricians and anesthetists. Maternal baseline demographic data such as age, geographic origin and body mass index (BMI) were collected in the medical and electronic charts. Clinical data related to medical, gynecological and obstetric history, pathologies and treatments during pregnancy, labor, delivery and neonatal care were also collected. Upon entry into the delivery room, a blood sample was collected for each woman to obtain a blood count, an exploration of coagulation (prothrombin time ratio (PT), aPTT ratio, fibrinogen and D-dimers) and the fraction of immature platelets (IPF).

All variables were reported as categorical variables in order to facilitate the use and interpretation of the care support tool by clinicians. The geographical origin was classified into five classes: Europe, Africa, Asia, overseas departments and territories and others. The gestational weight gain was categorized into three classes, according to US guidelines: adequate, excessive and insufficient weight gain [9]. Macrosomia was defined as a birth weight > 4000 g. Pre-pregnancy BMI was classified into four categories: underweight for a BMI < 18.5 kg/m^2^, normal for a BMI between 18.5 and 24.9 kg/m^2^, overweight for a BMI between 25 and 29.9 kg/m^2^ and obese for a BMI ≥ 30 kg/m^2^.

Antepartum bleeding included first trimester spotting as well as bleeding occurring later during the course of pregnancy and before delivery. Abnormal placentation was defined as one of the three following types of placental insertion: accreta, bipartita or previa. For biological variables, discretization thresholds were determined according to known cut-off used in the literature or, when no threshold was known, according to the optimal thresholds obtained using receiver operating characteristic (ROC) curves (area under the curve—AUC).

### 2.3. Statistical Analysis

#### 2.3.1. Description

The frequency of PPH was calculated in the entire population of the “HPP-IPF” derivation study and according to the mode of delivery—vaginal or cesarean section (C-section). Then, only vaginal deliveries were analyzed. To describe the study population, continuous variables were summarized as medians with interquartile ranges (IQR) and categorical variables as counts and percentages in each category. Incompleteness analysis (pattern, repartition, distribution and localization in dataset) was fully investigated, and missing data were reported as counts and percentages.

#### 2.3.2. Combination of Multiple Imputation and Variable Selection with Bootstrap

Multiple-chained equation imputation (MICE) was used to address missing values from candidate predictor variables. Multiple imputation takes into account uncertainty due to the presence of these missing data. In order to avoid additional noise to estimates, a minimal correlation of 0.2 between the variable being imputed and predictors being included in the imputation model was set. A number of imputations of 10 was also set to obtain precise and optimal estimates [7,10]. The method of imputation was chosen according to the type of variable. Quantitative variables were imputed using the predictive mean matching (PMM) approach. These variables were categorized only after imputation. Qualitative variables were imputed using logistic regression.

A selection of variables with bootstrap was carried out to determine which parameters associated with PPH had to be included in the final model [7]. This process helped to limit the excess of optimism in the predictive ability and fit of the model. This selection process was applied to 200 bootstrap samples on the 10 data sets previously imputed. In other words, 2000 samples were finally simulated, on which an automatic backward stepwise selection based on the Akaike criterion (AIC) was performed. The average of inclusion frequencies of each variable, i.e., the number of times they were retained in simulated models, was calculated. Variables retained in at least 80% of simulated models were included in the final multivariable model.

Thus, the combination of multiple imputation and the selection of variables with bootstrap allowed considering the variations due to imputation and sampling.

#### 2.3.3. Predictive Model, Score and External Validation

A multivariable logistic regression model was constructed to evaluate the ability of the clinical and biological parameters to predict PPH after vaginal delivery. To obtain the estimates of the coefficients and the standard deviations of the final model from the 10 imputed datasets, results were combined according to the rules established by Rubin [11]. The odds ratio (OR’s) and their 95% confidence interval (95% CI) were calculated. Performances of the predictive model were studied in terms of discrimination, with the c-index and the c-index corrected by bootstrap, and of calibration, with the slope of the prognostic index corrected by bootstrap [7,12]. Calibration refers to whether the predicted probabilities agree with the observed probabilities. The calibration slope is the estimated regression coefficient in a logistic regression model with the score as the only covariate [13].

Then, a predictive tool for PPH was built from the final logistic model in the form of a score, easy to use by clinicians. The choice of the threshold to classify women at risk for PPH from this predictive score was determined according to Youden’s optimal cut-off from the ROC curve.

An external validation of the predictive PPH score on the prospective “HEMOTHEPP” independent cohort was then performed, in terms of area under the curve, sensitivity and specificity, positive predictive value and negative predictive value.

Statistical significance was considered with an alpha risk of 5%. All statistical tests were performed using R version 4.0.4 software (USA).

## 3. Results

### 3.1. Descriptive Analysis

Between April 2013 and May 2015, 4162 deliveries occurred in the obstetrics unit of Brest University Hospital (HPP-IPF derivation cohort study). Among them, 563 refused to participate, 45 were excluded (birth before 21 weeks of gestation, a missing value for PPH or mode of delivery unavailable) and 812 were C-section (Figure 1). Finally, 2742 vaginal deliveries were included in this present study. Among those, PPH occurred in 141 (5.1%) women.

All clinical and biological characteristics of the overall population and according to the presence or absence of PPH are summarized in Table 1. The women’s median (IQR) age was 30 (26.0–33.0) years, with a median (IQR) BMI of 22.1 (20.1–25.7) kg/m². A total of 2221 (88%) women came from Europe, against 150 (6%) from Africa, 81 (3%) from the French overseas departments and territories and 24 (1%) from Asia. Few women had a medical or gynecological history, ranging from 0.2% for cardiovascular history to 9.6% for infection history. Among the 1612 women who had already given birth, 155 (9.6%) had a history of C-section and 64 (4.0%) had a history of PPH. A total of 124 (4.6%) pregnancies were assisted and 63 (2.3%) were multiple pregnancies. A total of 53 (1.9%) women presented an abnormal placental insertion (accreta, bipartita or previa). The pathologies observed during pregnancy were mainly gestational diabetes (11.3%), antepartum bleeding (7.2%) and premature delivery threat (5.8%). The median (IQR) labor duration was five (3.0–7.0) hours, with an induction of labor in 26% of cases. Labor was induced mainly (99%) with oxytocin. Macrosomia was observed in 189 (6.9%) newborns. An episiotomy was performed for 762 (27.8%) deliveries. Retained placenta occurred in 226 (8.2%) women. Some characteristics seemed more frequent in women who experienced PPH, such as a history of C-section or PPH, multiple pregnancies, excessive gestational weight gain, pre-eclampsia, antepartum bleeding, induction of labor, labor duration ≥ 8 h, episiotomy and platelets < 150 Giga/L.

The percentage of missing data ranged from 0% to 16.5% among the candidate predictor variables. The mean (SD) percentage of missing data was 3.9% (5.74): 0.9% for demographic and clinical characteristics and 11.9% for biological parameters.

### 3.2. Predictive Model

The selection of variables with bootstrap identified seven variables with a mean frequency of more than 80% (Appendix A): aPTT ratio ≥ 1.1, antepartum bleeding, pre-eclampsia, platelets < 150 Giga/L, labor duration ≥ 8 h, multiple pregnancy and macrosomia. These five clinical parameters and two biological parameters were included in the multivariable regression model.

The adjusted ORs (aOR) of these variables in the final model, their 95% CI and their *p*-value are presented in Table 2. All parameters were independently and significantly associated with PPH in the final regression model. The risk for PPH was increased with pre-eclampsia (aOR 6.41, 95% CI 2.47–16.65), antepartum bleeding (aOR 2.50, 95% CI 1.52–4.11), multiple pregnancy (aOR 3.15, 95% CI 1.49–6.65), labor duration ≥ 8 h (aOR 2.30, 95% CI 1.56–3.38), macrosomia (aOR 2.33, 95% CI 1.36–3.99), platelets < 150 Giga/L (aOR 2.45, 95% CI 1.40–4.30) and aPTT ratio ≥ 1.1 (aOR 1.96, 95% CI 1.22–3.13).

The AUC of the final predictive model of PPH was 0.68 (95% CI 0.63–0.72), and the AUC corrected by bootstrap, considering the over-optimism, was 0.67 (95% CI 0.63–0.71) (Figure 2). The slope of the prognostic index was 0.94, signifying a good agreement between the observed and predicted PPH.

### 3.3. Predictive Score

A predictive score, presented in Table 3, was then constructed from the final model by weighting based on the estimated coefficients. This score, ranging from 0 to 10 for each woman, was rounded to the unit in order to facilitate its use in routine practice. A high score indicates a high risk for PPH. The AUC of this score was 0.68 (95% CI 0.63–0.72) (Appendix A). An optimal threshold of 0.5 was determined using the ROC score curve in order to better identify all women at risk for PPH, i.e., sensibility was preferred. This threshold, corresponding to 0.5 point on a scale from 0 to 10, amounts to saying that as soon as a woman presents one of the seven characteristics of the predictive score, she is classified as at risk for PPH.

### 3.4. External Validation

This predictive score was then externally validated on the “HEMOTHEPP” cohort. Among 3061 vaginal deliveries, for which the biological results were available solely at Brest University Hospital, 218 (7.1%) PPHs occurred. The AUC of the score in this validation cohort was 0.63 (95% CI 0.59–0.65) (Appendix A). The score allowed identifying 64.7% of the women who had experienced PPH (a sensitivity of 64.7%). In other words, 35.3% of women who had experienced PPH were not detected by the score. Moreover, about 60% of women were predicted to be at risk for PPH among those who did not develop PPH (specificity of 59.0%).

In addition, about 11% of women had PPH among those who scored positive because they had at least one risk factor for PPH (a positive predictive value of 10.8%) and about 96% of women did not have PPH among those who scored negative (a negative predictive value of 95.6%).

## 4. Discussion

### 4.1. Main Findings

This study has enabled the development of a new predictive score for PPH in women giving birth vaginally from maternal clinical and biological data collected prospectively in a large derivation prospective cohort of women included at Brest University Hospital. This score was then validated externally in another large prospective cohort from the same hospital.

### 4.2. Strengths and Limitations

The main strength of this study is the in-depth statistical methodology used to construct this predictive model. Indeed, the combination of multiple imputation and selection of variables with bootstrap allowed considering the variations due to missing data and those due to sampling. This method turns out to be more efficient for developing a predictive model than using only one of these sources of variation [7]. The dataset contained approximately, on average, 4% missing data. Thus, multiple-chained equation imputation (MICE) was performed, creating 10 complete datasets. Multiple imputation considers the uncertainty due to the presence of missing values while preserving the structure of the data. It also allowed our model to perform better in terms of the precision of the parameter estimates, rather than only analyzing women with complete data. Then, on each complete dataset, a variable selection process with bootstrap was performed. Among all these simulated datasets, variables selected in at least 80% of simulated models were included in the final predictive model. This selection process with bootstrap allowed correcting the excess of optimism compared to the classical methods of automatic forward or backward stepwise selection. Austin suggests selecting variables with an inclusion frequency greater than 60% to obtain a model with a good predictive ability [14]. Our choice to fix the inclusion threshold at 80% allowed the selection of fewer variables for the predictive score in order to facilitate the use of this tool in practice, in this case, seven parameters.

The main weakness of our study is the choice of the threshold to classify pregnant women based on the score. This cut-off was set at 0.5 on a scale of 0 to 10 to ensure good sensitivity. This implies that as soon as a woman presents one of the seven characteristics of the predictive score, she will be considered at risk for PPH. A preventive therapeutic strategy adapted to the individual risk can still be proposed based on this score or at least allows the classification of women in different risk groups. An alternative to this score can be to calculate the exact probability for PPH using an automatic spreadsheet (Appendix A), as proposed by Rubio-Álvarez et al. [4].

Another limit of our study is that the score was validated only in women delivering in Brest University Hospital. One perspective would be to validate this predictive score on all the Finistère’s centers of the “HEMOTHEPP” cohort study (five other centers) and, thus, on different levels of maternity in order to have a representative sample of the French population. In addition, this score is not scalable. Indeed, the score includes both data related to maternal history, data on the ongoing pregnancy and data related to the childbirth in progress, collected at different times, until entry into the delivery room. It cannot, therefore, be used to sort out pregnant women before delivery, at any stage of pregnancy, and to refer the women at risk for PPH to the most appropriate maternity unit.

### 4.3. Interpretation

PPH, defined as a blood loss ≥ 500 mL in the 24 h following delivery, occurred in around 5% of the vaginal deliveries in our study population. This prevalence is consistent with that observed in several studies, including a previous study carried out in 106 French maternity hospitals between December 2004 and November 2006 [15].

Our predictive model has good discrimination and calibration, with an AUC of 0.68 in the derivation cohort and 0.63 in the validation cohort. These results are close, in terms of prediction, to other previous studies [2,3]. However, some of these models have not been validated [2,3]. The model of Rubio-Álvarez et al. had excellent discrimination abilities on the derivation and validation sample (AUC: 0.90 and 0.83), but data were collected retrospectively, and the authors did not use a consensual definition of PPH (reduction in hemoglobin levels greater than 3.5 g/dL in the 24 h following delivery) [4]. Several other studies aimed to predict severe PPH, defined as a blood loss > 1000 mL (which is the American definition of PPH but not the definition adopted by the World Health Organization and used in France or in other European countries [15,16,17,18,19,20,21,22]), and seem to have better predictive abilities with this outcome definition. In addition, some of these studies did not distinguish between vaginal and cesarean deliveries [16,17,20,21,22,23]. The construction of these predictive models is, therefore, carried out in a general childbirth population with a higher frequency of PPH due to the inclusion of deliveries by C-section. Predictive models are also often built from retrospective studies [4,17,19,21,23], sources of bias and confusion. Machine learning [16,18] would also seem to increase predictive abilities. Kartiz et al. used random forest and extreme gradient boosting to predict severe PPH (blood loss > 1000 mL) and found good discrimination (AUC = 0.93 and AUC = 0.62, respectively). However, this method has a major drawback: the clinical interpretation is difficult in practice.

In our model, five clinical factors (pre-eclampsia, antepartum bleeding, multiple pregnancy, macrosomia and labor duration ≥ 8 h) and two hematological factors (platelets < 150 Giga/L and aPTT ratio ≥ 1.1) were independently identified as associated with PPH. Most of these clinical parameters are known risk factors for PPH, validated many times in the literature [2,3,4,15,24,25]. Regarding the biological data predictive for PPH, the literature is poor. In most cases, thrombocytopenia, affecting about 10% of delivering women, is mild (between 100 and 149 Giga/L), benign and without risk of bleeding [26]. However, it can be associated with an underlying pathology such as pre-eclampsia. Along with this, mild thrombocytopenia has been recently described in an American monocentric cohort of term singleton deliveries as associated with severe PPH [27]. The aPTT ratio has been sparsely studied as a potential predictor for PPH. To our knowledge, an aPTT ratio ≥ 1.1 has not been previously described as associated with PPH. Only one previous study determined that aPTT in absolute value (≥38 s) was associated with PPH [28].

Some clinical parameters associated with PPH, but occurring too close to delivery, such as episiotomy and placental retention, were excluded from the analysis because they could not really predict the risk for PPH. Some other known PPH risk factors, such as maternal age, overweight/obesity, history of PPH, geographic origin, hemoglobin level or parity, were not selected either in our model [2,4,15,24]. In addition, some biological parameters not explored in previous studies, such as immature platelet fraction or fibrin monomers, were analyzed in this work while finally not selected.

This score could help to determine if a woman is at risk for PPH upon entry into the delivery room and then help to take preventive actions, in particular, the prophylactic administration of tranexamic acid. Indeed, tranexamic acid was proven effective in reducing maternal deaths and hysterectomies consecutive to PPH, without significant adverse effects, in the international multicenter randomized double-blind placebo-controlled WOMAN trial [29]. In 2018, TRAAP1, a French multicenter randomized trial, failed to demonstrate a reduction in PPH incidence after vaginal delivery in women receiving prophylactic tranexamic acid in addition to oxytocin [30]. The results of this trial may have been different if prophylactic tranexamic acid had been evaluated only in women at risk for PPH, identified before delivery, with a score similar to ours.

## 5. Conclusions

This work allowed developing an easy-to-use predictive score for PPH based on clinical and biological data collected prospectively from 2742 women who gave birth vaginally at Brest University Hospital. The methodology combining multiple imputation and bootstrap seems efficient, and the results are consistent with PPH risk factors previously identified in the literature. The validation of this score on an independent cohort, with acceptable discriminatory abilities, is also a strength of this work. However, this score needs to be validated in obstetrics units of different levels before being used routinely. Furthermore, it could also be implemented in future clinical trials evaluating the efficacy of tranexamic acid or other procedures on PPH prevention in women delivering vaginally.

## Figures and Tables

**Figure 1 biology-12-00054-f001:**
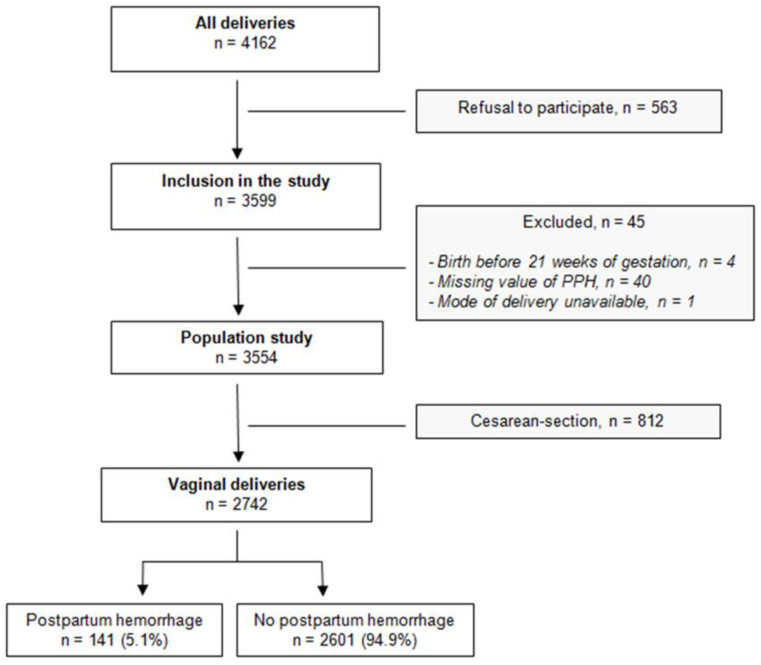
Flowchart of the derivation monocenter cohort.

**Figure 2 biology-12-00054-f002:**
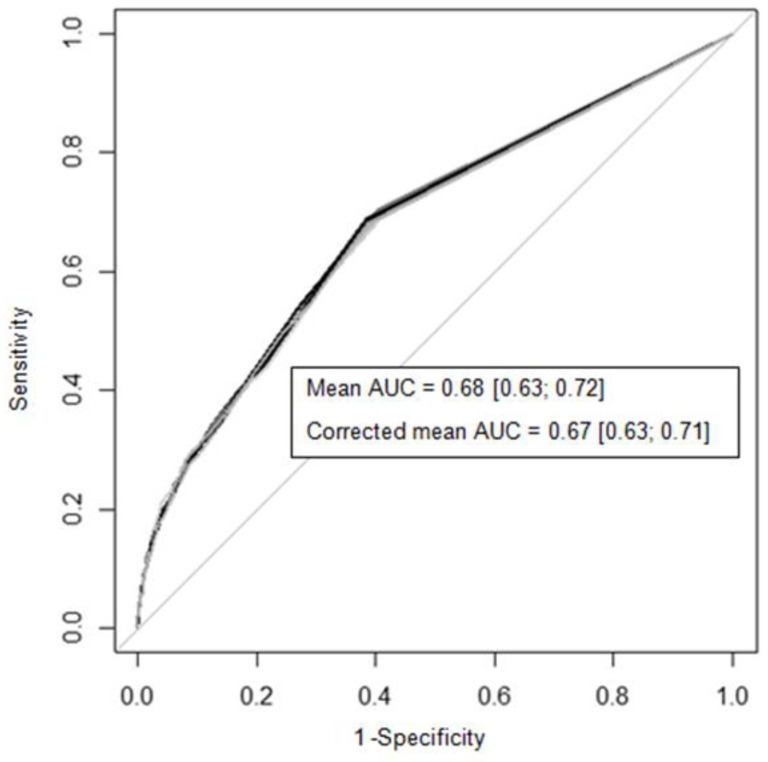
ROC curves of predictive model of PPH (bootstrap simulated samples). AUC—area under the ROC curve.

**Table 1 biology-12-00054-t001:** Clinical and biological parameters of women with vaginal delivery.

Variables	Overall * (*n* = 2742)	No PPH * (*n* = 2601)	PPH * (*n* = 141)
Maternal characteristics			
Age (years)	30.0 (26.0, 33.0)	30.0 (26.0, 33.0)	31.0 (26.0, 34.0)
Age < 35 years	2229 (81.3%)	2123 (81.6%)	106 (75.2%)
Age ≥ 35 years	513 (18.7%)	478 (18.4%)	35 (24.8%)
Body mass index (kg/m^2^)	22.1 (20.1, 25.7)	22.1 (20.1, 25.6)	22.9 (20.1, 27.3)
Underweight	230 (8.4%)	221 (8.5%)	9 (6.4%)
Normal	1729 (63.2%)	1645 (63.4%)	84 (59.6%)
Overweight	483 (17.7%)	460 (17.7%)	23 (16.3%)
Obese	293 (10.7%)	268 (10.3%)	25 (17.7%)
Ethnicity			
Europe	2221 (88.1%)	2112 (88.5%)	109 (82.0%)
Africa	150 (6.0%)	143 (6.0%)	7 (5.3%)
Asia	24 (1.0%)	21 (0.9%)	3 (2.3%)
Overseas departments and territories	81 (3.2%)	71 (3.0%)	10 (7.5%)
Others	44 (1.7%)	40 (1.7%)	4 (3.0%)
Medical history			
Bleeding history	38 (1.4%)	35 (1.3%)	3 (2.1%)
Cardiac disease	31 (1.1%)	29 (1.1%)	2 (1.4%)
Arterial disease	6 (0.2%)	5 (0.2%)	1 (0.7%)
Diabetes mellitus	18 (0.7%)	17 (0.7%)	1 (0.7%)
Infectious disease	262 (9.6%)	254 (9.8%)	8 (5.7%)
Venous thromboembolism	37 (1.4%)	35 (1.3%)	2 (1.4%)
Nephrological disease	24 (0.9%)	22 (0.8%)	2 (1.4%)
Transfusion history	60 (2.2%)	55 (2.1%)	5 (3.5%)
Autoimmune disease	31 (1.1%)	28 (1.1%)	3 (2.1%)
Gynecological history			
Bleeding history	18 (0.7%)	17 (0.7%)	1 (0.7%)
Uterine myoma	12 (0.4%)	10 (0.4%)	2 (1.4%)
Obstetric history			
Previous C-section **	155 (9.6%)	143 (9.3%)	12 (16.7%)
Previous PPH **	64 (4.0%)	57 (3.7%)	7 (9.7%)
Ongoing pregnancy			
Assisted reproductive technology	125 (4.6%)	109 (4.2%)	16 (11.4%)
Multiple pregnancy	63 (2.3%)	53 (2.0%)	10 (7.1%)
Parity			
0	1130 (41.2%)	1061 (40.8%)	69 (48.9%)
1 or 2	1417 (51.7%)	1356 (52.1%)	61 (43.3%)
≥3	195 (7.1%)	184 (7.1%)	11 (7.8%)
Weight gain			
Adequate	862 (33.7%)	823 (33.9%)	39 (30.5%)
Excessive	906 (35.4%)	849 (35.0%)	57 (44.5%)
Insufficient	789 (30.9%)	757 (31.2%)	32 (25.0%)
Smoking during pregnancy	677 (24.8%)	648 (25.0%)	29 (20.7%)
Alcohol during pregnancy	37 (1.4%)	35 (1.4%)	2 (1.4%)
Placenta previa, accreta, percreta	53 (1.9%)	47 (1.8%)	6 (4.3%)
Pathological outcomes during pregnancy			
Gestational hypertension	17 (0.6%)	15 (0.6%)	2 (1.4%)
Pre-eclampsia	27 (1.0%)	20 (0.8%)	7 (5.0%)
Gestational diabetes	310 (11.3%)	289 (11.1%)	21 (14.9%)
Premature delivery threat	160 (5.8%)	151 (5.8%)	9 (6.4%)
Antepartum bleeding	198 (7.2%)	176 (6.8%)	22 (15.6%)
Intrauterine growth restriction	25 (0.9%)	24 (0.9%)	1 (0.7%)
Premature rupture of membranes	67 (2.4%)	66 (2.5%)	1 (0.7%)
Hydramnios	12 (0.4%)	11 (0.4%)	1 (0.7%)
Intrahepatic cholestasis	32 (1.2%)	29 (1.1%)	3 (2.1%)
Treatments during pregnancy			
Anticoagulants	60 (2.2%)	58 (2.2%)	2 (1.4%)
Antiplatelets	63 (2.3%)	59 (2.3%)	4 (2.8%)
Anti-inflammatory drugs	27 (1.0%)	23 (0.9%)	4 (2.8%)
Psychiatric drugs	22 (0.8%)	21 (0.8%)	1 (0.7%)
Labor			
Labor induction	713 (26.0%)	661 (25.4%)	52 (36.9%)
Anesthesia			
None	421 (15.4%)	406 (15.6%)	15 (10.9%)
Epidural	2294 (83.9%)	2174 (83.7%)	120 (87.6%)
Spinal or general anesthesia	18 (0.7%)	16 (0.6%)	2 (1.5%)
Total labor duration (hours)	5.00 (3.00, 7.00)	5.00 (3.00, 6.54)	6.00 (4.00, 8.00)
Second stage of labor duration (minutes)	41 (15–102)	40 (15–101)	67 (20–110)
Delivery			
Temperature > 38 °C	6 (0.3%)	5 (0.2%)	1 (0.9%)
Macrosomia ***	189 (6.9%)	171 (6.6%)	18 (12.8%)
Instrumental birth	507 (18.5%)	474 (18.2%)	33 (23.4%)
Term of delivery (weeks of gestation)			
<37	250 (9.1%)	237 (9.1%)	13 (9.2%)
(37; 41)	2194 (80.1%)	2093 (80.6%)	101 (71.6%)
>41	295 (10.8%)	268 (10.3%)	27 (19.1%)
Vaginal lacerations	1043 (38.1%)	997 (38.4%)	46 (32.6%)
Episiotomy	762 (27.8%)	697 (26.8%)	65 (46.1%)
Retained placenta	226 (8.2%)	151 (5.8%)	75 (53.2%)
Biological parameters at admission in the delivery room			
Blood group O	1244 (45.4%)	1179 (45.3%)	65 (46.1%)
Hemoglobin (g/dL)	12.30 (11.50, 13.00)	12.30 (11.50, 13.00)	12.20 (11.30, 12.85)
Hematocrit (%)	36.10 (34.30, 37.90)	36.10 (34.30, 37.90)	35.70 (34.00, 37.80)
Platelets (Giga/L)	229 (194, 273)	230 (195, 274)	210 (174, 252)
Prothrombin time (%)	100.0 (94.0, 100.0)	100.0 (94.0, 100.0)	98.0 (92.0, 100.0)
aPTT ratio	1.00 (0.94, 1.06)	1.00 (0.94, 1.06)	1.01 (0.96, 1.07)
Fibrinogen (g/L)	5.08 (4.57, 5.68)	5.09 (4.58, 5.68)	4.89 (4.43, 5.60)
D-Dimers (µg/mL)	1.58 (1.15, 2.12)	1.57 (1.14, 2.11)	1.87 (1.31, 2.34)
Fibrin monomers (µg/mL)	5 (4, 8)	5 (4, 8)	6 (4, 8)
Immature platelet fraction (%)	5.0 (3.3, 7.5)	5.0 (3.3, 7.5)	5.2 (3.3, 8.0)
Mean corpuscular volume (fL)	87.2 (83.8, 90.4)	87.3 (83.8, 90.4)	86.7 (83.4, 90.2)
White blood cells (G/L)	11.2 (9.4, 13.6)	11.3 (9.4, 13.6)	10.7 (9.0, 12.6)
Neutrophils (G/L)	8.16 (6.48, 10.24)	8.18 (6.49, 10.25)	7.43 (6.11, 9.70)
Lymphocytes (G/L)	2.03 (1.64, 2.49)	2.04 (1.64, 2.49)	1.85 (1.56, 2.36)
Monocytes (G/L)	0.79 (0.64, 0.97)	0.79 (0.64, 0.97)	0.75 (0.60, 0.92)

* Median (IQR—inter-quartile range) for quantitative variables; *n* (%) for qualitative variables; ** in women with at least one previous delivery; *** macrosomia was defined as a birth weight > 4000 g; PPH—postpartum hemorrhage; C-section—cesarean section; and aPTT—activated partial thromboplastin time.

**Table 2 biology-12-00054-t002:** Multivariable regression model.

Variable	Adjusted OR	95% CI	*p*-Value *
Clinical parameters			
Pre-eclampsia	6.41	[2.47–16.65]	<0.001
Antepartum bleeding	2.50	[1.52–4.11]	<0.001
Multiple pregnancy	3.15	[1.49–6.65]	0.003
Labor duration ≥ 8 h	2.30	[1.56–3.38]	<0.001
Macrosomia **	2.33	[1.36–3.99]	0.002
Biological parameters			
Platelets < 150 Giga/L	2.45	[1.40–4.30]	0.002
aPTT ratio ≥ 1.1	1.96	[1.22–3.13]	0.005

* Wald test; ** macrosomia was defined as a birth weight > 4000 g; OR—Odds ratio; CI—confidence interval; and aPTT—activated partial thromboplastin time.

**Table 3 biology-12-00054-t003:** Predictive score for PPH.

Characteristics	Coefficients *	Modalities	Score
Pre-eclampsia	1.83	No	+0
		Yes	+3
Antepartum bleeding	0.91	No	+0
		Yes	+1
Multiple pregnancy	1.15	No	+0
		Yes	+2
Labor duration	0.83	<8 h	+0
		≥8 h	+1
Macrosomia **	0.86	No	+0
		Yes	+1
Platelets	0.90	≥150 Giga/L	+0
		<150 Giga/L	+1
aPTT ratio	0.67	<1.1	+0
		≥1.1	+1
**Maximum total score**			**+10**

* Estimated coefficients from the multivariable regression model; ** macrosomia was defined as a birth weight > 4000 g; and aPTT—activated partial thromboplastin time.

## Data Availability

The data that support the findings of this study are available from the corresponding author (CDM), upon reasonable request.

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
