# Peer review of "Development and Validation of a Predictive Tool for Postpartum Hemorrhage after Vaginal Delivery: A Prospective Cohort Study"

_biology, 2022, doi:10.3390/biology12010054_

Round 1
Reviewer 1 Report
Dear authors,
The manuscript is very interesting. A predictive score for PPH could be of great interest in current practice and I await the results of your future studies on the subject. However, the majority of the clinical parameters included in the predictive score are only qualitative and could be difficult to interpret in clinical practice. For example antepartum bleeding - many patients present minimum antepartum bleeding in the first trimester. Should they be included in the calculation of the score, or only patients with bleeding upon admision in the delivery room? The same question is for abnormal placentation (praevia or acreta). I consider a more detailed explanation of parameters useful.
Also line 374 - the number 8 is missing (labour duration> 8h)
Author Response
Dear reviewer, I would like to thank you very much for your kind words and your suggestions.
In our study, antepartum bleeding included first trimester spotting as well as bleeding occurring later during the course of pregnancy and before delivery. This definition is given on line 110 of the manuscript. So all these types of bleeding should be included in the calculation of the score.
Abnormal placentation was defined as one of the three following types of placental insertion: accreta, bipartita or previa (line 184). To be clearer, a definition of this parameter was added on line 111.
Finally, we added the number “8” forgotten on line 374.
Reviewer 2 Report
The present manuscript evaluates the validation and development of a possible predictive model for PPH after vaginal deliveries. The work is original, clear and might be useful in the obstetric setting. I therefore support the publication of this material but the presentation may be improved. The details are given below.
- why the authors decided to exclude from the analysis (inclusion criteria of the protocol) the PPH derived from a cesarean section ?
- regarding clinical parameters of women with vaginal deliveries, it is necessary include in the analysis the methods of induction, the second stage of labour duration, the obstetrics lacerations, and the cases of uterine atony.
Author Response
Dear reviewer, I would like to thank you very much for your comments.
- We chose to include only women with a vaginal delivery because pregnant women who deliver vaginally don’t have the same characteristics as pregnant women who deliver by C-section (women delivered by C-section have more comorbidities, have more often pregnancy complications and more often abnormal placental insertion). Therefore, they don’t have the same risk factors for PPH, even if they share some. They also don’t have the same level of risk for PPH (women delivered by emergency C-section have a much higher PPH risk than women delivered vaginally). On the other hand, while pregnant women delivered vaginally have a lower PPH risk, they are probably less carefully monitored than women delivered by C-section. Thus, when PPH happens, these women could be less well managed than women delivered by C-section. A predictive score mixing different modes of delivery would not reflect these disparities. For all those reasons, we chose to focus on pregnant women delivered vaginally.
- Regarding clinical parameters, we did not choose to take into account the method of induction, the duration of second stage of labour, and the obstetric lacerations. Vaginal lacerations come very late in the delivery process, so we considered it could not be evaluated as a “predictive” parameter for PPH. In the same way, we did not take into account episiotomy. We added these clinical data in the Table 1 detailing pregnant women’s characteristics. In the cases of labour induction, labour was induced mainly (99% of cases) with oxytocin. So we did not detail this parameter. We added this information in the results section on line 191. Finally, we did not specifically evaluate the effect of the second stage of labour duration on PPH risk, but we choose to evaluate the total duration of labour. We added the information on the second stage of labour duration in Table 1.
- Regarding uterine atony, we did not choose to study this parameter. Uterine atony is an etiology of PPH and not a risk factor for PPH. It is strongly associated with other parameters such as multiple pregnancy, macrosomia, length of labour…, so it is taken into account in some way in our score.
Reviewer 3 Report
The work should be considered one of the proposals for tools that are increasing the accuracy of PPH prediction. Its novelty lies in the development of a simple and transparent PPH risk scale based on a bootstrap. However, as the authors themselves write, the predictive score they developed has a sensitivity of 64.7% and specificity of 59%. Therefore, it can only be relatively useful - about as much as a large obstetrician's experience.
Several comments come to mind:
1/ Please justify why 21 weeks of pregnancy was used as the lower limit?
2/ The work applies to relatively young pregnant women - would it also apply to older ones, which is now a growing phenomenon? (could be included in the discussion)
3/ In the paper, the average weight is the same in pregnant women with and without PPH. Nevertheless, it would be appropriate to divide pregnant women into underweight, overweight and normal weight groups. The degree of nutrition may be relevant here. In addition, the social conditions of pregnant women may also matter, as suggested by the slightly higher rate of PPH in Asian women and pregnant women from overseas departments and territories compared to European women.
4/ Pregnancies after assisted reproductive techniques appear to be associated with higher rates of PPH.
5/ In constructing the bootstrap, adenomyosis was not considered while taking into account blood group or eosinophil percentage, for example, is unnecessary. (Table 1 can be simplified)
6/ Table 3 is redundant, the text is sufficient.
7/Str. 5, row 172 and 173: "(Figure 1)" is enough to put once at the end of row 174.
Author Response
Dear reviewer, I would like to thank you very much for your comments.
1/ Births before 21 WG were excluded in this study. However, as mentioned in the flow chart (figure 1), only 4 deliveries have been excluded due to this exclusion criterion. So it did not induce a high selection bias.
2/ We cannot extrapolate if our score would have the same performances for PPH prediction in a population of older pregnant women.
In HPP-IPF cohort study, maternal median age was 30.00 (26.00-33.00) years, and 18.7% women were ≥ 35 years old. In this cohort, maternal age dichotomized as < 35 or ≥ 35 years (information added in Table 1) did not appear as a strong risk factor for PPH and, as a consequence, was not selected int the final model with the bootstrap method.
In the external validation cohort, i.e. women included in HEMOTHEPP study, median age of participants was a little bit higher (30.75 (27.51-34.29) years) than in HPP-IPF study, and 21% women were ≥ 35 years old.
3/ Maternal weight was classified according to the 2009 IOM guidelines for gain weight during pregnancy: < 18.5, [18.5-25[, [25-30[, ≥ 30 kg/m2. This definition was added in the method section on line 110. We also added the information on BMI according to these classes in the Table 1.
We were not able to evaluate the social conditions of the participants, but we agree it should have an impact on pregnancy morbidity in general, and on PPH in particular.
4/ You are right, assisted conception is a known risk factor for PPH, but this parameter was not selected in the final model with the bootstrap method.
5/ Thank you for this remark. We retrieved the eosinophil count and the basophil count from Table 1.
6/ Thank you for this remark. We retrieved the Table 3 from the manuscript.
7/ Thank you for this remark. We retrieved the second “Figure 1” at the end of the sentence.
Reviewer 4 Report
The authors present a study where they try to develop a predictive tool for postpartum hemorrhage after vaginal delivery. My main concerns are:
1. The graphical presentation of the results should be improved. This will make it more accessible to the reader.
2. Line 374: The sentence does not include the duration of labor. The sentence should be corrected.
3. There are few grammatical and lexical errors in the manuscript. Authors should pay more attention to the entire text and make necessary corrections
Author Response
Dear reviewer, I would like to thank you very much for your comments.
- We are sorry, but we did not find out how to improve the graphical presentation of the results, since we have mainly tables with data, which are not really “graphical”. We would be grateful if you could give us some suggestions to improve it, if it is really needed.
- Thank you for this remark. The sentence was modified and the number “8” was added.
- Thank you for this remark. We re-read and corrected the whole manuscript and hope there will not be any error left.